

# Modelling the deep convective transport of trace gases (CO, NH₃ and SO₂) from the planetary boundary layer to the Asian summer monsoon anticyclone

Jianzhong Ma[1,2,3], Bin Chen[1], Qianshan He[4], Xiaolu Yan[1,2], Gaili Wang[1,3,5], Siyang Cheng[1,2,3], Benedikt
Steil[6], Christoph Brühl[6], Andrea Pozzer[6], and Jos Lelieveld[6]

[1]Institute of Tibetan Plateau Meteorology & State Key Laboratory of Severe Weather Meteorological Science and Technology, Chinese Academy of Meteorological Sciences, Beijing 100081, China
[2]Heavy Rain and Drought-Flood Disasters in Plateau and Basin Key Laboratory of Sichuan Province, Institute of Tibetan Plateau Meteorology, China Meteorological Administration, Chengdu 610213, China
[3]Mêdog National Climate Observatory & Mêdog Field Research Station for Atmospheric Water Cycle, China Meteorological Administration, Linzhi 860700, China
[4]Shanghai Meteorological Service & Shanghai Key Laboratory of Meteorology and Health, Shanghai 201199, China
[5]Mêdog Atmospheric Water Cycle Observation and Research Station of Xizang Autonomous Region, Linzhi, 860700, China
[6]Atmospheric Chemistry Department, Max Planck Institute for Chemistry, Mainz, Germany

*Correspondence to*: Jianzhong Ma (majz@cma.gov.cn)

**Abstract.** Deep convection plays a vital role in transporting Asian pollutants from the planetary boundary layer (PBL) into the Asian summer monsoon anticyclone (ASMA). However, the efficiency and effectiveness of transporting pollutants with various chemical and physical properties to the ASMA remain unclear. In this study, we use the global atmospheric chemistry and climate model EMAC to investigate the deep convective transport of trace gases such as CO, NH₃ and SO₂ from the PBL 20 to the ASMA over the years 2010-2020. We quantify the deep convective transport efficiency of different trace gases into the ASMA. We show that the strongest convective transport tendency occurs over northern India and the southern edge of the Tibetan Plateau for CO (0.2-0.5 ppbv hr$^{-1}$), over the south and eastern parts of the Tibetan Plateau for NH₃ (0.02-0.05 ppbv hr$^{-1}$), and over central India and eastern China for SO₂ (0.002-0.005 ppbv hr$^{-1}$). We find that, in contrast to CO and NH₃, the SO₂ enhancements within the ASMA are very weak, and there is can even be a decrease in SO₂ over the southern Tibetan 25 Plateau relative to the surroundings. Our analysis indicates that gas-liquid partitioning in clouds and subsequent wet deposition over South Asia are more effective at reducing SO₂ than NH₃ reaching the Tibetan Plateau and the ASMA. In view of ongoing changes in regional emissions, the effects of deep convective transport of various pollutants and associated gas-aerosol-cloud interactions on the chemical features of the ASMA require continued investigation.



## 1 Introduction

Deep convection plays an important role in the vertical redistribution of trace gases and the pollution transport from the planetary boundary layer (PBL) to the upper troposphere and lower stratosphere (UTLS) (Chatfield and Crutzen, 1984;Dickerson et al., 1987;Lelieveld and Crutzen, 1994). The rapid transport of reactive pollutants from the PBL to the free troposphere and the tropopause region can significantly impact the UTLS chemistry globally (Pickering et al., 1990;Thompson et al., 1994;Thornton et al., 1997;Barth et al., 2007;Bertram et al., 2007;Barth et al., 2012). The deep and widespread convection associated with the Asian summer monsoon is a convective manifestation observed in June, July, and August that is absent in the Northern Hemisphere wintertime (Houze Jr. et al., 2015). Deep convection can transport Asian pollutants, especially those from South Asia, to the Asian summer monsoon anticyclone (ASMA) (Hoskins and Rodwell, 1995), where they are confined by the ASMA flow, forming a distinct chemical regime in the UTLS of the Northern Hemisphere during summertime (Randel and Park, 2006;Park et al., 2007;Park et al., 2008;Randel et al., 2010). The strength of the barrier separating air masses inside and outside the anticyclone depends on altitude, and there are also daily and yearly variations of the ASMA in relation to its boundary, area, and dynamic behaviour (Ploeger et al., 2015;Kachula et al., 2025). Satellite observations have revealed increasing levels of tropospheric gases (e.g., CO, HCN and $NH_3$) within the ASMA relative to outside it (Kar et al., 2004;Randel et al., 2010;Höpfner et al., 2016;Santee et al., 2017). A widespread strong enhancement of aerosols within the ASMA, namely the Asian Tropopause Aerosol Layer (ATAL), has also been detected by satellites (Vernier et al., 2011;Thomason and Vernier, 2013;Vernier et al., 2015).

The ASMA can serve as "an efficient smokestack" venting Asian surface pollutants to the global stratosphere (Yu et al., 2017;Bian et al., 2020). While the central role of the ASMA is recognized, there is still inconsistency about the most efficient pathway for the transport of moisture and air pollutants to the UTLS (Chen et al., 2012;Chen et al., 2024). The Tibetan Plateau plays a critical role in transporting air pollutants from the lower troposphere to the stratosphere during the Asian summer monsoon (Zhou et al., 1995). Deep convection over the Tibetan Plateau and its southern slope is considered "a short circuit" for the transport of water vapor and polluted air to the global stratosphere (Li et al., 2005;Fu et al., 2006). The middle troposphere centered over the southern Tibetan Plateau can act as "a well-defined conduit", where strong convection lofts air parcels in the boundary layer into the ASMA (Bergman et al., 2013;Bian et al., 2020). In addition, the northern Bay of Bengal and adjacent land area, where air pollution from the Indian subcontinent converges, has been identified as the central aerosol-convection source area for the ATAL (e.g., Park et al., 2009;Fadnavis et al., 2013;He et al., 2021;Nützel et al., 2022). The emissions from South Asia have been shown to make a dominant contribution to gaseous and aerosol pollutants trapped in the ASMA (Bergman et al., 2013;Lelieveld et al., 2018;Ma et al., 2022). While $SO_2$ and $NO_2$ emissions in India have been observed to increase rapidly (Krotkov et al., 2016), how these South Asian pollutants are transported to the ASMA and how they affect its chemical features remains an area in need of in-depth investigation.

The ASMA constitutes a seasonally persistent, zonally restricted circulation pattern transporting climate-relevant emissions rapidly from surface sources to higher altitudes (Vogel et al., 2015;Vogel et al., 2016;Ploeger et al., 2017;Vogel et al., 2023).



Transport by deep convection generally occurs alongside scavenging and multiphase chemical reactions involving reactive trace gases and aerosols, and it is also accompanied by various gas-aerosol-cloud interactions (Iribarne and Pyshnov, 1990;Pruppacher and Klett, 1997;Zondlo et al., 1997;Seinfeld and Pandis, 2006;Bertram et al., 2007). In addition to the convective dynamic activity, the deep convective transport efficiency of air pollutants, which reflects the amount of gases or
aerosols reaching the outflow at the cloud top relative to that of the inflow into the convective mass near the cloud base, can change significantly with the difference in their solubility, hydrophilicity and reactivity (Barth et al., 2007;Barth et al., 2015;Yang et al., 2015;Bela et al., 2016). Therefore, understanding the various physical and chemical processes involved in the deep convective transport is essential for revealing the chemical characteristics of air masses within the ASMA and the formation mechanism of the ATAL. Several measurement campaigns have been conducted successfully to explore the
chemical composition and spatiotemporal distributions of air pollutants in the UTLS over the Asian summer monsoon region, including the High Altitude and Long Range Research Aircraft (HALO) measurement (e.g., Gottschaldt et al., 2017), the Oxidation Mechanism Observations (OMO) campaign (Lelieveld et al., 2018;Hottmann et al., 2020), the Balloon Measurements of the Asian Tropopause Aerosol Layer (BATAL) (Vernier et al., 2018;Vernier et al., 2022), the StratoClim field campaign (Höpfner et al., 2019;Johansson et al., 2020;Lee et al., 2021;von Hobe et al., 2021;Appel et al., 2022;Singer et
al., 2022;Ebert et al., 2024;Johansson et al., 2024), the Asian Summer Monsoon Chemical and CLimate Impact Project (ACCLIP) (e.g., Smith et al., 2025), and the Probing High Latitude Export of Air from the Asian Summer Monsoon (PHILEAS) campaign (Riese et al., 2025;Vogel et al., 2025). However, these measurements in Asia have still not fully characterized and quantified the deep convective transport of air pollutants from the PBL to the ASMA, compared to the Deep Convective Clouds and Chemistry (DC3) field experiment taking place in North America (e.g., Barth et al., 2015).

In the present study, we use the ECHAM/MESSy Atmospheric Chemistry (EMAC) model (Jöckel et al., 2006;Jöckel et al., 2010) to investigate the deep convective transport of trace gases, including carbon monoxide (CO), ammonia ($NH_3$) and sulfur dioxide ($SO_2$), from the PBL to the ASMA. CO is a typical pollution tracer that is transported into and trapped within the ASMA (Fu et al., 2006;Park et al., 2007). $NH_3$ and $SO_2$ have a much higher solubility and reactivity in water clouds than CO, and can act as gasesous precursors of ammonium sulfate aerosols (Seinfeld and Pandis, 2006). Therefore, deep convective
transport of $NH_3$ and $SO_2$ from the PBL to the ASMA may significantly influence the formation processes and chemical composition of the ATAL (Höpfner et al., 2019). There is substantial interannual variability in the ASMA. For example, in summer 2015, the monsoon, in particular upward transport in the ASMA, was strongly influenced by El Niño, which has a global impact lasting over many months (Kunze et al., 2016;Santoso et al., 2017;Yan et al., 2018;Fadnavis et al., 2019;Ravindra Babu et al., 2021;Becker et al., 2025). Further, the composition of the ASMA tends to have strong day-to-day variability,
influenced by various meteorological conditions such as typhoons (Vogel et al., 2014;Hanumanthu et al., 2020;Li et al., 2020;Li et al., 2021;Li et al., 2023). For this study, we performed model simulations using EMAC for a relatively long period, from January 2010 through December 2020. This study aims to investigate the deep convective transport of selected trace gases to the ASMA for a climatology of 2010-2020.





This paper is organized as follows. In Sect. 2, we describe the EMAC model and settings used for this study. In Sect. 3, we present the model simulation results, including the tendency and efficiency of the deep convective transport of CO, NH$_3$ and SO$_2$ into the ASMA. In Sect. 4, we provide a discussion exploring the effects of partitioning into the clouds and wet scavenging processes on the deep convective transport of NH$_3$ and SO$_2$ into the ASMA. A summary of the main findings and conclusions

is given in Sect. 5.

## 2 Model description and setup

The EMAC model is a global atmospheric chemistry and climate model that combines the 5th generation European Centre – Hamburg general circulation model (ECHAM5) (Roeckner et al., 2006) with the Modular Earth Submodel System (MESSy) Atmospheric Chemistry system (Jöckel et al., 2006;Jöckel et al., 2010) to simulate atmospheric processes from the troposphere

to the middle atmosphere and their interactions with oceans, land, and anthropogenic influences. For this study, we used the EMAC (ECHAM5 version 5.3.02, MESSy version 2.55.0), which includes the MESSy submodels describing various chemical, physical and dynamical processes in detail (Jöckel et al., 2016). The model resolution used in this study is T63L90, corresponding to a horizontal grid resolution of about 1.875$^o$ $\times$ 1.875$^o$ and 90 vertical layers extending from the Earth's surface to an altitude of 0.01 hPa (~80 km).

In EMAC, heterogeneous and gas-phase chemistry are simulated online using the MECCA submodel (Sander et al., 2011;Sander et al., 2019). MECCA calculates the concentration of a range of gases and radicals, including reactive and aerosol precursor species, such as CO, SO$_2$, NH$_3$, nitrogen oxides (NO$_x$ $\equiv$ NO +NO$_2$), and volatile organic compounds (VOCs), and major oxidant species like OH, O$_3$, H$_2$O$_2$, and NO$_3$. In this study, the Mainz isoprene mechanism (Taraborrelli et al., 2009), and halogen and sulfur stratospheric chemistry (Brühl et al., 2015), were included in the MECCA calculation, following our

previous work (Ma et al., 2019). Photolysis rates for the troposphere up to the mesosphere are calculated by the JVAL submodel (Jöckel et al., 2006), which considers absorption and scattering by gases, aerosols and clouds using a delta-two-stream method. The uptake of SO$_2$ and NH$_3$ and the aqueous-phase oxidation of SO$_2$ by H$_2$O$_2$ and O$_3$ in cloud droplets are calculated by the SCAV submodel (Tost et al., 2006a;Tost et al., 2007). The removal of gases and aerosols through wet deposition is calculated by the SCAV submodel (Tost et al., 2006a), and dry deposition is calculated by the DRYDEP submodel (Kerkweg et al., 2006).

Aerosol microphysics and gas/aerosol partitioning are treated by the GMXe submodel (Pringle et al., 2010), which uses seven interacting lognormal modes (M7) to describe the typical size range of aerosol species, including the nucleation mode and hydrophilic and hydrophobic Aitken, accumulation and coarse modes. The properties of aerosols in each mode are represented by the number concentration, total mass (internal mixture of contributing species), density, median radius, and width of the lognormal distribution. The inorganic aerosol composition is simulated by the ISORROPIA-II thermodynamic

equilibrium model (Fountoukis and Nenes, 2007), with updates as discussed in the work of Capps et al. (2012).

Convective cloud processes and convective tracer transport are calculated using the CONVECT and CVTRANS submodels (Tost et al., 2006b), respectively. CONVECT consists of an interface for selecting different convection schemes (Tost et al.,



2006b), and in this study we used the Tiedtke convection scheme with Nordeng closure (Tiedtke, 1989;Nordeng, 1994), which has proven to be the best-performing, including in the recent work of Xenofontos et al. (2025). Convective cloud microphysics is based on temperature and moisture profiles, though without interactively accounting for the influence of aerosols on liquid droplet or ice formation processes. Large-scale cloud processes are described by the CLOUD submodel (Roeckner et al., 2006),
5 and the original ECHAM5 cloud scheme without aerosol–cloud interactions was used for this study.

The CAMS-GLOB-ANT emission inventory (v4.2, https://eccad.sedoo.fr, last access: 1 November 2025), with a horizontal grid resolution of 0.5 °×0.5 °at monthly intervals, was utilized for surface anthropogenic emissions of the trace gases (including CO, $NH_3$ and $SO_2$) and aerosols in this study. For aircraft emissions, the CAMS-GLOB-AIR emission inventory (v1.1, https://eccad.sedoo.fr, last access: 1 November 2025) was used. The land and water biological emissions of $NH_3$ and non-
10 methane hydrocarbons (NMHCs) are based on the Global Emissions Inventory Activity (GEIA) database (Bouwman et al., 1997). $NO_x$ produced by lightning is calculated online and distributed vertically based on the parameterization of Price and Rind (1992). The NO soil emissions are calculated online according to the algorithm of Yienger and Levy II (1995). Biomass burning emissions are calculated by the BIOBURN submodel (Kaiser et al., 2012), which determines the fluxes based on biomass burning emission factors and dry matter combustion rates from the Global Fire Assimilation System (GFAS).

The $SO_2$ emissions from explosive volcanic eruptions occurring at different locations and latitudes were taken into account using the volcanic $SO_2$ plumes detected by various (occultation and limb-based) satellite instruments (Schallock et al., 2023). There were about 146 explosive volcanic eruption events accounted for the years 2010-2020, and a three-dimensional volcanic $SO_2$ plume (in unit of volume mixing ratio) for each event was added on-line to the background values of $SO_2$ in the UTLS at the corresponding time during the model simulation, as done in our previous work (Brühl et al., 2018;Ma et al., 2019).

The EMAC model has been evaluated against various observations of trace gases and aerosols in both the troposphere and stratosphere, including aircraft measurements conducted within the ASMA in some specific years (Gottschaldt et al., 2017;Lelieveld et al., 2018;Tomsche et al., 2019;Johansson et al., 2020;Xenofontos et al., 2024). These comparisons indicate that EMAC can simulate complex dynamic, physical and chemical processes from the PBL to the UTLS over the Asian summer monsoon region (e.g., Gottschaldt et al., 2018;Ma et al., 2019;Rosanka et al., 2021;Becker et al., 2025). In this study, the model
simulation was performed for the years 2010–2020 at an integration time step of 10 minutes. The simulation was nudged by the ECMWF's ERA5 meteorological re-analysis data at time intervals of 6 hours (Hersbach et al., 2020). The nudging was exerted for temperature, vorticity, divergence, and surface pressure, with maximum weights at the model levels from about 10 hPa to 706 hPa (except for surface pressure). The chemical initial conditions of trace gases and aerosols were provided by the results of a previous simulation using EMAC T106L90 (Ma et al., 2019). The simulation results were output at 5-hour intervals
for analysis.

## 3 Results

The objective of this study is to investigate the deep convective transport of the primary gaseous pollutants CO, $NH_3$ and $SO_2$ from the planetary boundary layer (PBL) to the Asian summer monsoon anticyclone (ASMA). During the summer season, i.e.,





June-August (JJA), deep convection associated with the Asian summer monsoon is observed each year (Houze Jr. et al., 2015) and is selected for the analysis. We focus on the climatology of deep convective transport of selected trace gases into the ASMA in JJA during 2010-2020.

### 3.1 Spatial distributions of CO, NH₃ and SO₂ in the PBL during the Asian summer monsoon

The Asian summer monsoon region is characterized by enhanced precipitation, strong and deep convection, and a distinct vertical structure in the circulation, with a cyclonic flow and convergence in the lower troposphere and a strong anticyclone and divergence in the UTLS (Krishnamurti and Bhalme, 1976;Hoskins and Rodwell, 1995;Wang and LinHo, 2002). These climatic features associated with the Asian summer monsoon are well captured by our EMAC model with 11 years of simulated meteorological data (see Fig. S1). Figure 1 shows the averaged PBL's wind field and volume mixing ratios of CO, NH₃ and

SO₂ in JJA over the years 2010-2020. The influence and footprints of the large anthropogenic emission sources near the Earth's surface, especially those in northern India and eastern China (see Fig. S2), are clearly evident in the geographic distributions of these primary gaseous pollutants within the PBL. As shown in Fig. 1, in the PBL, the pollutants from South Asia (northern India in particular) can be transported by the cyclonic flow (dominated by southwesterlies over the Indian subcontinent) to the southern flank of the Tibetan Plateau. Further transport of these pollutants to the large Tibetan Plateau platform appears to be

limited by the topographical block at the steep southern slopes of the plateau with less thermal impact (Boos and Kuang, 2010).

While the PBL's CO levels are comparable between northern India and eastern China, the PBL's NH₃ levels are 5-10 ppbv higher in northern India than in eastern China and, in contrast, the PBL's SO₂ levels are 4-8 ppbv lower in northern India than in eastern China (Fig. 1). For NH₃, IASI satellite observations also showed a similar spatial distribution pattern, with higher abundance over northern India than eastern China, due to stronger emission strength from the former (Liu et al., 2022;Luo et

al., 2022). As can be estimated from Fig. 1, the ratios of the PBL's NH₃ volume mixing ratio to the PBL's SO₂ volume mixing ratio are about 4-6 in northern India, and these ratios drop to 1.5-2 in eastern China. Since NH₃ is an alkaline gas and SO₂ is an acidic gas, changes in their relative abundance will affect aqueous-phase chemistry and the scavenging process when they are dissolved in clouds  (Seinfeld and Pandis, 2006).

### 3.2 Deep convection frequency and spatial distributions of CO, NH₃ and SO₂ within the ASMA

Deep convection usually occurs as part of a mesoscale convective system with a sufficient moisture supply at cloud base (Houze Jr., 2004). Strong ascent with deep convection can reach the upper troposphere above 10 km above sea level (a.s.l., hereafter, all altitudes are referred to a.s.l. unless specified otherwise), and even higher into the lower stratosphere (Meenu et al., 2010;Emmanuel et al., 2021). The ASMA is located at an altitude range of about 10 km to 18 km in the vertical, and horizontally it covers a larger area ranging 30-120°E and 20-40°E, which can be highlighted using the geopotential height

distribution at 100 mPa (Fig. S1) as done by Basha et al. (2020). For this study, we used the convective cloud top heights output at intervals of 5 hours over the simulation period to count the events of deep convection reaching a selected altitude



(e.g., 10 km) or higher. Then, we calculated the frequency of deep convection at the defined altitudes by dividing the number of deep convection events by the total number across all datasets (including both cloudy and cloud-free cases) used for counting.

Figure 2 presents the deep convection frequency for JJA during 2010-2020 for the convective cloud top heights above 10 km, 12 km, 14km, and 16 km, respectively, corresponding to the main vertical range of the ASMA. Except for the oceanic

regions, the frequency of deep convection reaching altitudes above 10 km is the highest (up to ~20%) over the Tibetan Plateau among the continental Asian summer monsoon regions. Even at altitudes above 14 km, the frequency of deep convection is prominently high (up to 10%) over the Tibetan Plateau compared to the surrounding regions within the ASMA. It is also shown that deep convection can reach to altitudes above 16 km over the Tibetan Plateau, although the corresponding frequencies are as small as ~1% or less, lower than those over the surrounding areas to its south within the ASMA. These results are in

agreement with previous studies in that convective clouds form frequently over the Tibetan Plateau due to the abundant water vapor convergence, elevated land surface and strong radiative heating (e.g., Sugimoto and Ueno, 2010;Xu et al., 2014). They are also consistent with the work of Legras and Bucci (2020), which shows that the Tibetan Plateau, with the largest number of high clouds, favours the deep convective transport of compounds released from ground level to much higher altitudes, even into the lower stratosphere.

Figure 3 shows the averaged volume mixing ratios of CO, $NH_3$ and $SO_2$ at the selected altitudes of 10-12 km and 14-16 km in JJA over the years 2010-2020, respectively. It can be seen that the spatial distributions of CO, $NH_3$, and $SO_2$ within the ASMA differ. The enhancements of CO and $NH_3$ within the ASMA are clearly visible, with the maxima occurring over northern India and the southern edge of the Tibetan Plateau for CO and over the southern and eastern parts of the Tibetan Plateau for $NH_3$. The CO spatial distribution character within the ASMA, as simulated in this study, has been well recognized

by a large number of previous studies (e.g., Park et al., 2007;Santee et al., 2017). The $NH_3$ enhancements within the ASMA characterized by this study are also in agreement with the satellite observations and model study before (Höpfner et al., 2016;Ge et al., 2018;Ma et al., 2019).

An analysis of the MIPAS satellite data showed the enhanced mixing ratios of $SO_2$ at altitudes of 16-18 km within the ASMA (Höpfner et al., 2015), but these enhancements of $SO_2$ are not as significant as those of CO and $NH_3$. Figure 3 indicates

that $SO_2$ enhancements occur over central India (a small area around 20 °N latitude) and eastern China, but these enhancements are very weak. In contrast, our simulation shows a decrease of $SO_2$ over the southern Tibetan Plateau relative to the surroundings within the ASMA, which was not detected by satellite measurements. According to our simulation, $SO_2$ levels within the ASMA are very low (below 0.05 ppbv), similar to those reported by Höpfner et al. (2015). For $SO_2$, the averaged mixing ratios in the upper troposphere within the ASMA are about two orders of magnitude lower than those in the polluted

area within the PBL, whereas they are about one order of magnitude lower for $NH_3$ and only a few times lower for CO.

## 3.3 Tendency and efficiency of the deep convective transport of CO, $NH_3$ and $SO_2$ into the ASMA

As mentioned in Sect. 2, convective transport of the tracers was simulated using the CVTRANS submodel, with the updraft and downdraft mass fluxes, entrainment and detrainment calculated by the CONVECT submodel. In the CVTRANS scheme,



convective transport is calculated separately from scavenging, and the tracers are redistributed vertically without a net gain or loss in the whole convective column. The tendency of a tracer due to convective transport alone can be obtained from the difference in the tracer's mixing ratio before and after the implementation of CVTRANS. Here, in this study, the mean tendency is calculated by averaging the data across all time intervals, rather than only for deep convection events.

Figure 4 shows the averaged deep convective transport tendency of CO, $NH_3$ and $SO_2$ at the selected altitudes of 10-12 km and 14-16 km in JJA over the years 2010-2020, respectively. Strong deep convective transport tendency within the ASMA can be found over northern India and the southern edge of the Tibetan Plateau for CO (0.2-0.5 ppbv $hr^{-1}$), over the southern and eastern parts of the Tibetan Plateau for $NH_3$ (0.02-0.05 ppbv $hr^{-1}$), and over central India and eastern China for $SO_2$ (0.002-0.005 ppbv $hr^{-1}$). These spatial distribution patterns in the deep convective transport tendency are similar to those in the volume

mixing ratios of CO, $NH_3$ and $SO_2$ shown in Fig. 3, indicating a dominant role of deep convection in the enhancements of CO, $NH_3$ and $SO_2$ within the ASMA. It can be seen that the maximum tendency for $SO_2$ (e.g., over central India) is much lower (by an order of magnitude) than that for $NH_3$ (e.g., over the southern Tibetan Plateau). Moreover, in contrast to CO and $NH_3$, an enhancement of the strong deep convective transport tendency cannot be found for $SO_2$ over the southern Tibetan Plateau. It is indicated that, in addition to the deep convective transport itself, other factors, e.g., the scavenging process associated with

it, influence the amount of pollutants reaching the ASMA.

In this study, the deep convective transport efficiency of a trace gas into the ASMA is defined by the ratio of the updraft mass flux of this trace gas at a selected height (i.e., 10 km) to its maximum near the cloud base in the convective column (see Fig. S3). The updraft mass flux of a trace gas is calculated as the updraft mass flux of the air times the mass mixing ratio of this trace gas at the same altitude. It can be expected that, in addition to convective activity, cloud scavenging may influence

the deep convective transport efficiency of the trace gas by reducing the amount that reaches the ASMA.

Figure 5 shows the mean deep convective transport efficiency of CO, $NH_3$ and $SO_2$ into the ASMA (defined with a lower boundary of 10 km height) in JJA over the years 2010-2020, respectively. The deep convective transport efficiency is very high (above 50%) over the central and southern Tibetan Plateau for all three trace gases considered, indicating that the middle troposphere of the Tibetan Plateau is an effective pathway for transporting pollutants from the PBL to the ASMA. With a low

solubility and thus not affected by scavenging, CO has a higher deep convective transport efficiency than $NH_3$ and $SO_2$ over all the continental regions where deep convection occurs. For instance, over the southern Tibetan Plateau, the maximum in the deep convective transport efficiency reaches about 80%-90% for CO, 60%-70% for $NH_3$, and 60%-80% for $SO_2$. Over northern India (excluding the southern flank of the Tibetan Plateau), the deep convective transport efficiency is 20%-30% for CO, 2%-4% for $NH_3$, and less than 1% for $SO_2$. The deep convective transport efficiency of $SO_2$ calculated here appears to be

comparable to that of $NH_3$ over the southern Tibetan Plateau It should be noted that there could be an overestimation of the deep convective transport efficiency of $SO_2$ here since a considerable fraction of $SO_2$ in the upper troposphere might come from other sources than Asian surface emissions, e.g., volcanic $SO_2$ from explosive eruptions (Neely et al., 2014;Ma et al., 2019).





## 4 Discussion

In this section, we aim to explain the results presented in the above section (Sect. 3) by investigating the cloud and wet scavenging processes over the Asian summer monsoon region during the same period (i.e., JJA during 2010-2020). Among the three trace gases, only the cloud and wet scavenging processes of $NH_3$ and $SO_2$ are investigated since CO has a low

solubility and is not considered in the EMAC submodel SCAV. The focus is on the causes of the difference between $NH_3$ and $SO_2$ in the deep convective transport from the PBL to the ASMA.

### 4.1 Partitioning of NH₃ and SO₂ between the gas and liquid phases in clouds

In the above section, we show that the middle troposphere over the Tibetan Plateau is an effective pathway of transporting the pollutants from the PBL to the ASMA by deep convection, with some differences in the deep convective transport efficiency

for different trace gases, e.g., CO vs. $NH_3$, and $NH_3$ vs. $SO_2$ (Fig. 5). To investigate the effect of cloud process on the differences, we looked at the vertical column densities of gaseous $NH_3$ and $SO_2$ in the air (denoted as $NH_3$(air) and $SO_2$(air), simply as $NH_3$ and $SO_2$), their dissolving and reaction products in the clouds (denoted as NHx(liq) and SOx(liq)), and the total (i.e., $NH_3$(air)+NHx(liq) and $SO_2$(air)+SOx(liq)) within a height range of 6-10 km for the simulation period (see Fig. S4). The NHx in the clouds is the sum of liquid ammonia and ammonium, i.e., NHx(liq) $\equiv$ $NH_3$(liq) + $NH_4^+$(liq). The SOx in the clouds is

the sum of liquid $SO_2$ and its dissociation and oxidation products (denoted by S(IV) and S(VI)), i.e., SOx(liq) $\equiv$ S(IV, liq) + S(VI, liq), where S(IV, liq) $\equiv$ $SO_2$(liq) + $HSO_3^-$(liq) + $SO_3^{2-}$(liq) and S(VI, liq) $\equiv$ $H_2SO_4$(liq) + $HSO_4^-$(liq) + $SO_4^{2-}$(liq). It is noted that for the investigated region and period of this study, NHx(liq) and SOx(liq) are dominated by $NH_4^+$(liq) (>99.9%) and $SO_4^{2-}$(liq) plus $HSO_4^-$(liq) (>99%), respectively.

Figure 6 shows the relative contributions of the liquid NHx and SOx in the clouds to the total (gas-phase plus liquid-phase)

within a vertical column of 6-10 km in JJA during 2010-2020, respectively. One can see considerable amounts of gaseous $NH_3$ and $SO_2$ at heights above 6 km over the Tibetan Plateau partitioning into the clouds before reaching the ASMA. Similar to the spatial distributions of $NH_3$ and $SO_2$, the enhancements in the 6-10 km vertical columns of cloudy NHx(liq) and SOx(liq) occur over the Tibetan Plateau as well (Fig. 4S). The relative contributions of the 6-10 km column in the clouds to the total (gas-phase plus liquid-phase) over the Tibetan Plateau are about 10-30% for NHx(liq) and 50-80% for SOx(liq), respectively. In

this study, the primary partitioning of the soluble gases between the gas and liquid phases was treated with Henry's law, with a Henry's law constant of 203 M atm$^{-1}$ for $NH_3$ and 3.13 M atm$^{-1}$ for $SO_2$ at a temperature of 0 ℃, respectively. While the environmental conditions, including the cloud liquid water content, are the same, the lower Henry's law constant for $SO_2$ than for $NH_3$ cannot explain the higher fraction of $SO_2$ than $NH_3$ partitioning into the liquid phase simulated by the model. The gases dissolved in water can dissociate into ions, e.g., $NH_4^+$(liq), $HSO_3^-$(liq), and $SO_3^{2-}$(liq), following chemical equilibria

included in the SCAV submodel. Therefore, the effective Henry's law coefficients, which consider trace gas dissociating into ions, are better to represent the partitioning of the trace gases like $NH_3$ between the gas and liquid phases (Seinfeld and Pandis, 2006). Under atmospheric conditions, practically all dissolved ammonia in clouds exists as the ammonium ion. Moreover, in





the clouds the concentration level of S(VI, liq) is two orders of magnitude higher than that of S(IV, liq) according to our simulation (not shown). This indicates that the oxidation of S(IV, liq) to S(VI, liq) is an important factor influencing the partitioning of $SO_2$ between the gas and liquid phases, and it can result in higher fractions of $SO_2$ than $NH_3$ partitioning into the liquid phase, as shown in Fig. 6.

## 4.2 Wet removal of $NH_3$ and $SO_2$ through aqueous $NH_4^+$ and S(VI) in the ASM region

Being very soluble and reactive in water, $NH_3$ and $SO_2$ partitioned into the cloud droplets can be efficiently removed from the atmosphere by wet deposition of their aqueous reaction products NHx(liq) and SOx(liq). As for NHx(liq) and SOx(liq) in the clouds mentioned above (Sect. 4.1), NHx(liq) and SOx(liq) in rain are also mainly (>99%) in the form of $NH_4^+$(liq) and $SO_4^{2-}$(liq) plus $HSO_4^-$(liq), respectively. Figure 7 shows the wet deposition fluxes of NHx(liq) and SOx(liq) averaged for JJA over

10 the years 2010-2020, respectively. Strong wet deposition fluxes of NHx(liq) and SOx(liq) occur in the polluted regions, such as northern India and eastern China, corresponding to the large emission rates of $NH_3$ and $SO_2$ there (Fig. S2). They are also affected by the precipitation rate and spatial distribution (Fig. S1c). For example, the simulated SOx(liq) wet deposition fluxes are negligible in Saudi Arabia due to extremely low precipitation, although the $SO_2$ emissions are large in some areas there. The maximum wet deposition fluxes of both NHx(liq) and SOx(liq) are found on the southern and eastern slopes of the Tibetan

Plateau, corresponding to the highest precipitation rates there.

Numerical studies have highlighted the very strong precipitation occurring along the southern slopes of the Tibetan Plateau and its role in the monsoon circulation (e.g., Bao and Li, 2020). This precipitation removes aerosols over the Tibetan Plateau by wet scavenging (Liu et al., 2023). Our model results indicate that in addition to the topographical block, the removal by wet scavenging along the southern slopes is very effective at reducing the amounts of $NH_3$ and $SO_2$ reaching the Tibetan Plateau

(Fig. 1 and Fig. 7). Such scavenging is more effective for $SO_2$ than $NH_3$, resulting much lower amounts of $SO_2$ available for the deep convective transport from the lower troposphere of the Tibetan Plateau to the ASMA. While the deep convection frequency (shown in Fig. 2) is a dominant factor, the concentration levels of the tracers in the PBL (shown in Fig. 1) and scavenging efficiency are also important for determining the amount of the tracers transported into the ASMA (Fig. 4). This can explain the causes of the minima in both the deep convective transport tendency and volume mixing ratio of $SO_2$ within

the ASMA occurring over the southern Tibetan Plateau.

We quantify the effectiveness of precipitation in removing $NH_3$ and $SO_2$ by comparing their atmospheric lifetimes with wet deposition, calculated as the tropospheric vertical column density (below 10 km) divided by the wet deposition flux for each trace gas. Figure 8 shows the mean atmospheric lifetime of tropospheric $NH_3$ and $SO_2$ against wet deposition in JJA during 2010-2020. It can be seen that over the southern parts and slopes of the Tibetan Plateau, the lifetime against wet

deposition is around 1-2 days for $NH_3$ and less than 1 day for $SO_2$, respectively. The lifetime of $SO_2$ is shorter than that of $NH_3$ against wet deposition over northern India, the Tibetan Plateau and its southern slopes, and vice versa over eastern China. Such a turnaround could be due to differences in the relative amounts of tropospheric $SO_2$ and $NH_3$ over these regions. The ratio of the tropospheric vertical column density of $NH_3$ to that of $SO_2$ appears to be much larger over northern India than over eastern





China (see Fig. S5). The $NH_3$ column over northern India is the largest over the globe as reported in a previous study (Wang et al., 2020). It should be noted that $NH_3$ and $SO_2$ are alkaline and acidic gases, tending to neutralize each other to form $NH_4^+$(liq) and $SO_4^{2-}$(liq) or $HSO_4^-$(liq) in the clouds, respectively. The large and excess amounts of $NH_3$ relative to $SO_2$ over northern India not only favour the deep convective transport of more $NH_3$ from the PBL to the ASMA, but also limit the amounts of $SO_2$ transported to the Tibetan Plateau by enhancing its partitioning into the clouds and removal by wet deposition.

## 5 Conclusions

We have investigated the deep convective transport of trace gases CO, $NH_3$ and $SO_2$ from the planetary boundary layer (PBL) to the Asian summer monsoon anticyclone (ASMA) using the global atmospheric chemistry and climate model EMAC at T63L90 resolution. The model simulation was performed for the period from January 2010 to December 2020, and for this study 11 years of the seasonally averaged results for June–August (JJA) have been analyzed. The focus is on the similarities and differences in the transport efficiency of CO, $NH_3$, and $SO_2$ from the PBL, in particular from the South Asian PBL, to the ASMA.

By analysing the spatial distributions of convective cloud top height, we show that the frequency of deep convection reaching into the ASMA (defined as having a lower boundary at 10 km) is the highest (up to 20%) over the Tibetan Plateau among the continental Asian summer monsoon regions. Such a high deep convection frequency favours the efficient transport of trace gases from the lower troposphere of the Tibetan Plateau to the ASMA. Here for the first time, we quantify the deep convective transport efficiency of the trace gases CO, $NH_3$ and $SO_2$ into the ASMA by the ratio of the updraft mass flux of trace gas at the lower boundary of the ASMA (i.e., 10 km height) to its maximum near the convective cloud base. The results show that the deep convective transport efficiency is very high (above 50%) over the Tibetan Plateau for all three trace gases considered, with maximum values of about 80%-90% for CO, 60%-70% for $NH_3$, and 60%-80% for $SO_2$. Over the Indo-Gangetic plane, the deep convective transport efficiency is 20%-30% for CO, 2%-4% for $NH_3$, and less than 1% for $SO_2$.

Our model simulations show that the strongest deep convective transport tendency within the ASMA tends to occur over northern India and the southern edge of the Tibetan Plateau for CO (0.2-0.5 ppbv $hr^{-1}$), over the southern and eastern parts of the Tibetan Plateau for $NH_3$ (0.02-0.05 ppbv $hr^{-1}$), and over central India and eastern China for $SO_2$ (0.002-0.005 ppbv $hr^{-1}$). These spatial distribution patterns of deep convective transport tendency match well with those of the volume mixing ratios of CO, $NH_3$ and $SO_2$ within the ASMA, respectively. While the enhancements of CO and $NH_3$ within the ASMA are evident, the enhancements of $SO_2$ are very weak. In contrast to CO and $NH_3$, we find a decrease in $SO_2$ over the southern Tibetan Plateau relative to the surroundings within the ASMA.

We find considerable amounts of gaseous $NH_3$ and $SO_2$ at heights above 6 km over the Tibetan Plateau partitioning into the clouds before reaching the ASMA, with the relative contributions of the 6-10 km column in the clouds (liquid-phase) to the total (gas-phase plus liquid-phase) of about 10-30% for $NH_3$ (in the form of $NH_4^+$(liq)) and 50-80% for $SO_2$ (mainly in the form of $SO_4^{2-}$(liq) and $HSO_4^-$(liq)), respectively. Our model simulations show that the removal by wet deposition on the southern slopes plays an important role in reducing the amounts of $NH_3$ and $SO_2$ reaching the Tibetan Plateau platform. Such

scavenging is more effective for $SO_2$ than for $NH_3$, with lifetimes of 1-2 days for $NH_3$ and less than 1 day for $SO_2$. We argue that the large and excess amounts of $NH_3$ relative to $SO_2$ over northern India not only favour the deep convective transport of more $NH_3$ from the PBL to the ASMA, but also limit the amounts of $SO_2$ transported to the Tibetan Plateau and then the ASMA by enhancing its partitioning into the clouds and removal by wet deposition. Moreover, deep convection is generally

accompanied by lightning activity, which can release $NO_x$ and further produce $HNO_3$ through oxidation reactions. While $NH_3$ is alkaline, $SO_2$ and $HNO_3$ are acidic gases, and they are all gaseous precursors of aerosols in the ATAL Ongoing changes in emissions of $NH_3$, $SO_2$ and $NO_x$ and their interactions through multi-phase chemistry during deep convective transport may have significant impacts on the chemical characteristics of ASMA, which require continued investigation.

**Data availability.** The usage of MESSy (Modular Earth Submodel System) and access to the source code is licensed to all

affiliates of institutions which are members of the MESSy Consortium. Institutions can become members of the MESSy Consortium by signing the "MESSy Memorandum of Understanding". More information can be found on the MESSy Consortium website: http://www.messy-interface.org (last access: 3 November 2025). The code used in this study has been based on MESSy version 2.55 and is archived with a restricted-access DOI (https://doi.org/10.5281/zenodo.8379120, The MESSy Consortium, 2023). The data produced in the study are available from the authors upon request.

**Supplement.** The supplement related to this article is available online.

**Author contributions.** JM, QH and JL initiated the project. JM performed the model simulations and analyzed the data. BC, QH, XY, GW, and SC contributed to the data analysis. BS, CB, AP, and JL contributed to the model simulation. JM prepared the manuscript with contributions from all co-authors.

**Competing interests.** At least one of the (co-)authors is a member of the editorial board of Atmospheric Chemistry and Physics.

The authors have no other competing interests to declare.

**Special issue statement.** This article is part of the special issue "The Modular Earth Submodel System (MESSy) (ACP/GMD inter-journal SI)". It is not associated with a conference.

**Acknowledgements.** JM would like to thank Patrick Jöckel, Rolf Sander, and other MESSy colleagues for their help in using EMAC and the submodels.



**Financial support.** This research has been supported by the National Natural Science Foundation of China (grant nos. 42330603, 42475123).

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




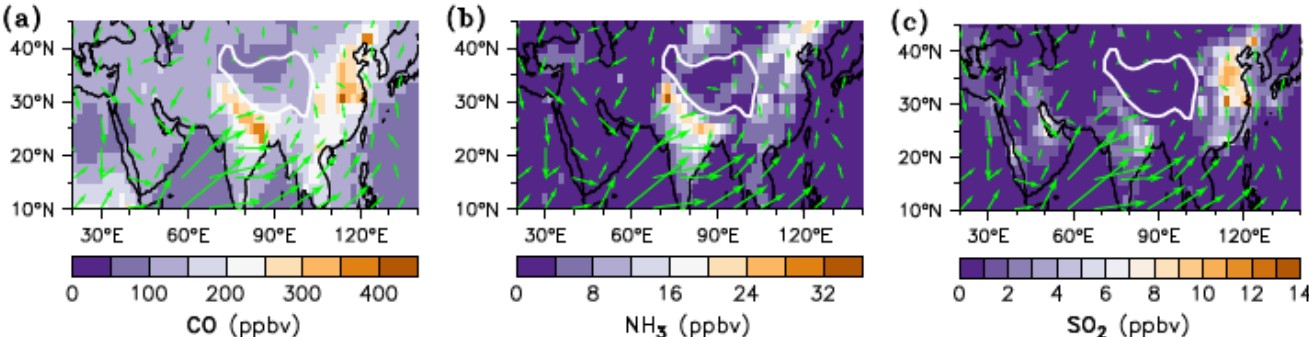

**Figure 1.** EMAC simulated volume mixing ratios of CO **(a)**, NH₃ **(b)** and SO₂ **(c)** overlaid with the wind vectors in the PBL, averaged for JJA over the years 2010-2020. White lines represent the 3 km terrain height contour, highlighting the Tibetan Plateau.

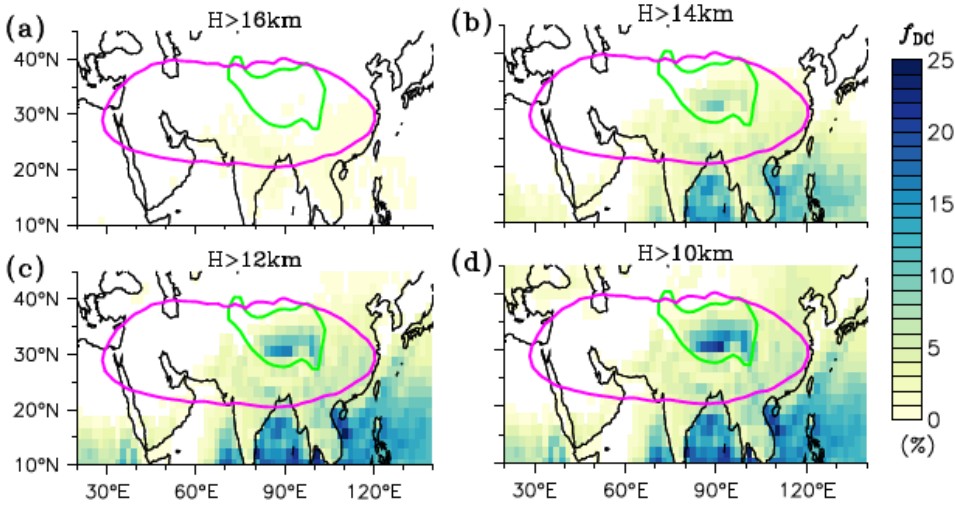

**Figure 2.** EMAC simulated relative deep convection frequency ($f_{DC}$ in percent) for selected convective cloud top heights, i.e., above 16 km **(a)**, 14 km **(b)** 12 km **(c)**, and 10 km **(d)** above sea level, in JJA during the years 2010-2020. Purple lines are the 16.64 km geopotential height contour at 100 hPa, highlighting the main ASMA area (see Figure S1). Green lines represent the 3 km terrain height contour, highlighting the Tibetan Plateau.



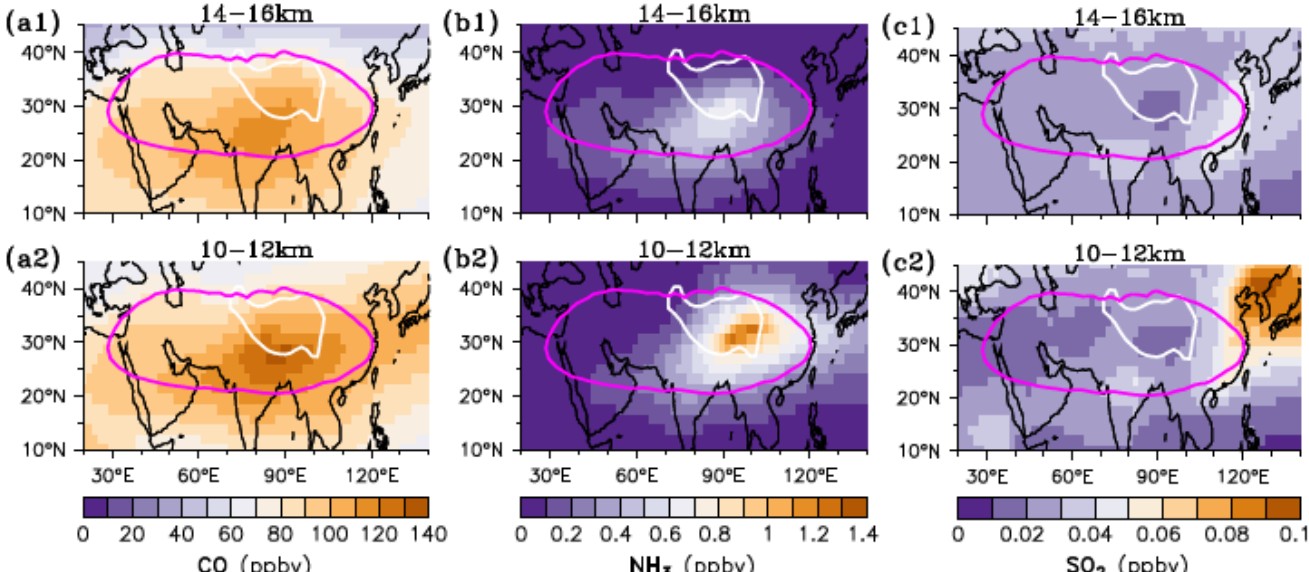

**Figure 3.** EMAC simulated volume mixing ratios of CO (**a1** and **a2**), NH₃ (**b1** and **b2**) and SO₂ (**c1** and **c2**) at selected altitudes, i.e., 14-16 km (**a1**, **b1** and **c1**) and 10-12 km (**a2**, **b2** and **c2**) above sea level, averaged for JJA over the years 2010-2020. Purple lines show the 16.64 km geopotential height contour at 100 hPa, highlighting the main ASMA area (see Figure S1). White lines represent the 3 km terrain height contour, highlighting the Tibetan Plateau.

.





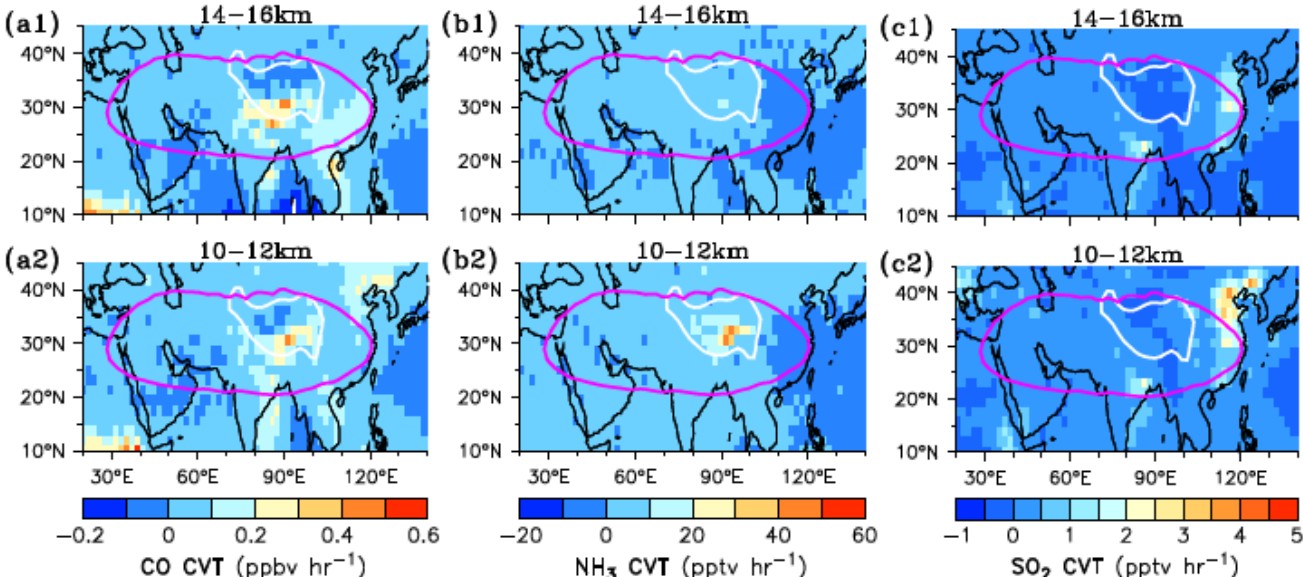

**Figure 4.** EMAC simulated mean deep convective transport tendency (CVT) of CO (**a1** and **a2**), NH$_3$ (**b1** and **b2**) and SO$_2$ (**c1** and **c2**) at selected altitudes, i.e., 14-16 km (**a1**, **b1** and **c1**) and 10-12 km (**a2**, **b2** and **c2**) above sea level, averaged for JJA over the years 2010-2020. Purple lines show the 16.64 km geopotential height contour at 100 hPa, highlighting the main ASMA area (see Figure S1). White lines represent the 3 km terrain height contour, highlighting the Tibetan Plateau.

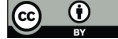



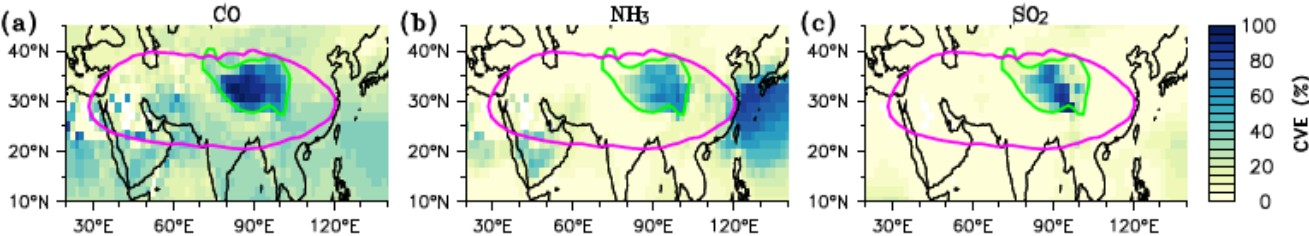

**Figure 5.** EMAC simulated mean deep convective transport efficiency (CVE), i.e., the ratio of the updraft mass flux (UMF) at 10 km height above sea level to its maximum in the deep convection column (expressed in percent), for CO **(a)**, $NH_3$ **(b)** and $SO_2$ **(c)**, in JJA during the years 2010-2020. Purple lines show the 16.64 km geopotential height contour at 100 hPa, highlighting the main ASMA area (see Figure S1). Green lines represent the 3 km terrain height contour, highlighting the Tibetan Plateau.

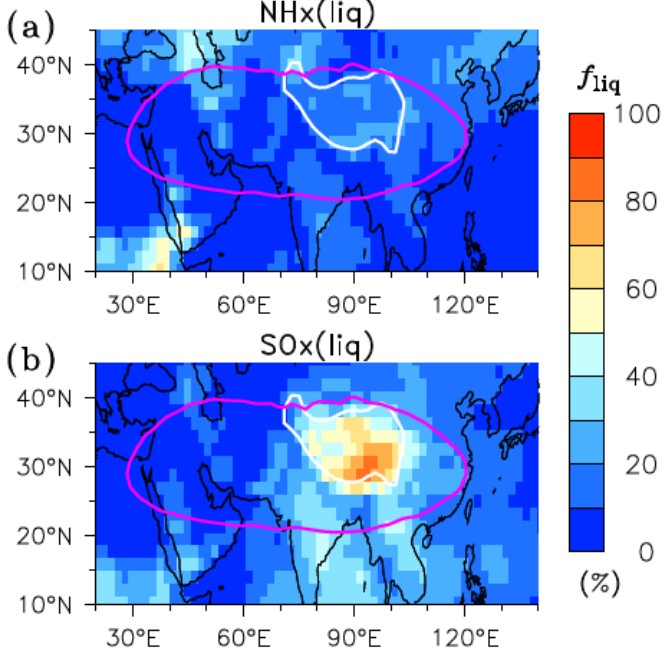

**Figure 6.** EMAC simulated mean relative contributions of the liquid-phase amount to the total (gas-phase plus liquid-phase) within a vertical column of 6-10 km above sea level ($f_{liq}$), for $NH_3$ and its reaction products in the clouds (denoted by NHx) **(a)** and $SO_2$ and its reaction products in the clouds (denoted by SOx) **(b)**, in JJA during the years 2010-2020. Purple lines show the 16.64 km geopotential height contour at 100 hPa, highlighting the main ASMA area (see Figure S1). White lines represent the 3 km terrain height contour, highlighting the Tibetan Plateau.





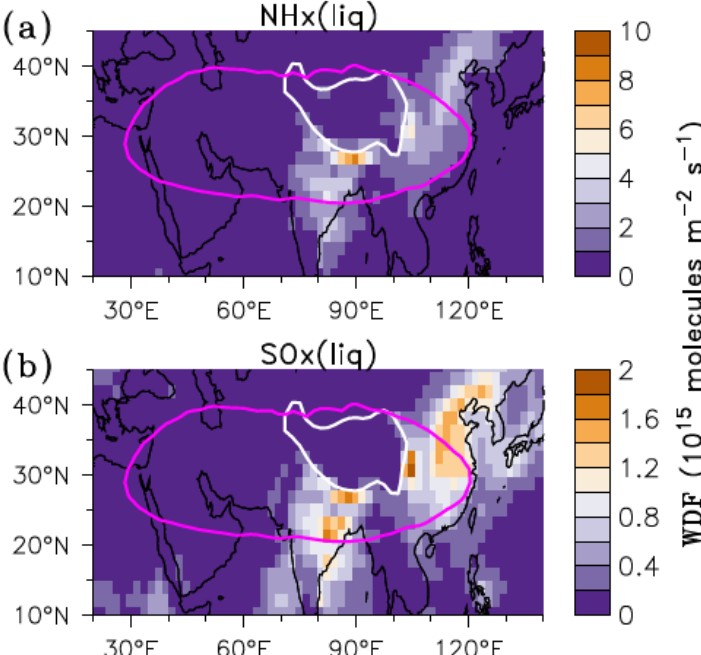

**Figure 7.** EMAC simulated wet deposition fluxes (WDF) of NH$_3$ and its reaction products in the rains (denoted by NHx) (**a**) and SO$_2$ and its reaction products in the rains (denoted by SOx) (**b**), averaged for JJA over the years 2010-2020. Purple lines show the 16.64 km geopotential height contour at 100 hPa, highlighting the main ASMA area (see Figure S1). White lines represent the 3 km terrain height contour, highlighting the Tibetan Plateau.





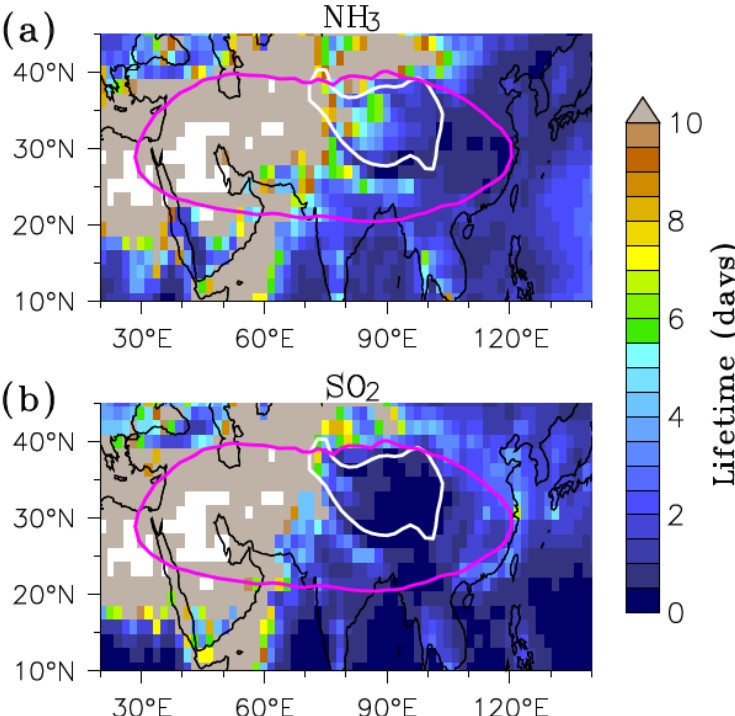

**Figure 8.** EMAC simulated mean atmospheric lifetimes of tropospheric NH$_3$ **(a)** and SO$_2$ **(b)** (below 10 km height above sea level) against wet deposition in JJA during the years 2010-2020. Purple lines show the 16.64 km geopotential height contour at 100 hPa, highlighting the main ASMA area (see Figure S1). White lines represent the 3 km terrain height contour, highlighting the Tibetan Plateau.