# Peer review of "Modelling the deep convective transport of trace gases (CO, NH3 and SO2) from the planetary boundary layer to the Asian summer monsoon anticyclone"

_EGUsphere, 2025_

## Referee Comment (RC1)

**Review of "Modelling the deep convective transport of trace gases (CO, NH3 and SO2) from the planetary boundary layer to the Asian summer monsoon anticyclone" by J. Ma et al.**

**Summary:** This study investigates the model representation of convective transport from the Asian boundary layer into the Asian summer monsoon anticyclone (ASMA) over the period 2010-2020.  Three trace gases are considered in the analysis. The examined simulation indicates that convective transport tendency is strongest over the Tibetan Plateau for CO and NH3, and over India and China for SO2.

**Overall Thoughts:** Overall this is a well-written study that makes an important contribution, and I believe that it should eventually be published. The authors include and discuss a large amount of relevant literature, which is a nice feature of the work. My primary concern is that I do not believe that the lone simulation presented can be trusted as representative of the real ASMA without some comparison to available observations. As the authors point out in the introduction, there have been a number of satellite, ground-based, and airborne measurements in the ASMA region, and I believe leveraging these for evaluation prior to detailed analysis of the model results is an important but currently missing feature.

**Recommendation:** Major revision

**General Remarks:**

- As mentioned above, the comprehensive literature review in this study is quite nice. One aspect that appears missing is the recent discovery of the importance of East Asian emissions for the composition of the ASMA (see for example Smith et al., 2025 already cited and Pan et al., 2024, https://doi.org/10.1073/pnas.2318716121). This seems point seems to be missing in the first Introduction paragraph.
- On page 3 the authors imply that the DC3 campaign has fully characterized and quantified deep convective transport, which I don't think any airborne campaign could truly do.  I suggest instead that this remark be reframed that airborne measurements alone, however valuable, are not sufficient to fully characterize and quantify deep convective transport due especially to limited sampling in space and time, thus necessitating the use of numerical modeling to provide such estimates. This can serve to nicely motivate the present study as well.

- The model representation of convective transport is integral to this study, but I don't see any mention or reference to how the CVTRANS submodel actually works. Moreover the cited Tost et al. (2006b) study does not seem to explicitly mention either the CONVECT or CVTRANS submodels. It would be helpful for the authors to provide some more information or documentation here for reader interest.
- Related to the above comment, it would be good if the authors could clarify whether convective transport tendency and convective transport efficiency calculations are performed within the model physics code immediately prior to and following convective transport. If not, I would expect that other processes would need to be considered too, such as horizontal transport from surrounding regions and chemical losses.
- The first three sentences in Section 3 seem unnecessary to me, as the information should have already been made clear in Sections 1 and 2. The middle sentence in particular doesn't make sense to me, as there doesn't appear to be any "observed" convection involved in this study, only simulated convection.
- It's worth noting that CAMS emissions of SO2 over China likely underestimate recent reductions due to environmental policies over 2010-2020, as described by an upcoming publication (https://doi.org/10.22541/essoar.175682819.92297398/v1). This might be the reason why SO2 convective transport is highest over China in the simulation presented herein. This could be mentioned as Figure 1c is described, as these model PBL values are mostly controlled by the emissions used as input.
- Are the results in Figure 2 supported by observations? It seems that convection is mostly concentrated over high terrain and it's important to verify that with observations if we want to believe the subsequent model results of convective transport tendency. Figure 1 of Smith et al (2025, already cited) does not show this same signal in satellite observations over the Tibetan Plateau, albeit for a single and different year.
- As suggested above, I think it is also important to compare the results presented in Figure 3 (volume mixing ratios) with available observations to demonstrate the simulation's realism. Otherwise it is hard to know whether the subsequent model results are likely to be representative of the real ASMA environment.
- I question how useful the convective transport efficiency calculation presented in Figure 5 is, as it seems to just highlight the same region where most of the model convection is (the Tibetan Plateau, Figure 2). Perhaps transport efficiency is high there because of the enhanced convection, but the Plateau is a region where emissions are comparatively low (Figure 1) so this efficient transport still shouldn't impact ASMA composition all that much.

- I don't believe that the title of "Discussion" for Section 4 is appropriate, I would suggest something related to model scavenging processes instead. It seems to me that this section is meant to provide additional analysis to give context to what was presented in Section 3. In my view a discussion is a reflection of past analysis rather than a presentation of new analysis.
- I believe that the conclusions section is missing some big-picture link(s) back to ASMA research as a whole. The early literature review is nice, and I think it would help to discuss how previous studies can be put into context by these targeted model results, or specifically how future research in this field should be guided. Compatibility of the simulation with observations will add critical confidence that the real ASMA behaves in a similar way to support such claims.

**Technical Remarks and Typos:**

- Page 1 Line 24: typo, remove "is".
- Page 2 Line 7: I recommend removing "a convective manifestation".
- Page 2 Line 15: consider replacing "outside it" with "its surroundings".
- Page 2 Line 18: I see that the term "smokestack" appears once in the Yu et al. (2017) study, but I think the "vertical conduit" terminology from Bergman et al. (2013) is much more widely accepted. I believe it should be used throughout this paragraph.
- Page 3 Line 4: I suggest removing "dynamic".
- Page 3 Line 16: The appropriate citation for the ACCLIP campaign is the recently published overview paper (Pan et al., 2025, https://doi.org/10.1029/2025JD044417).
- Page 6 Line 30: "mPa" should be "hPa".
- Page 7 Line 7: whether a 10% frequency is "high" is subjective. I would rephrase to just say that the frequency reaches 10%. The same thing occurs on Page 8 Line 23.
- Page 7 Line 16: I don't think that "respectively" is required in this case.
- Page 7 Line 21: I suggest replacing "model study before" with "past model studies".
- Page 8 Line 30: missing period.
- Page 12 line 6: missing period.
- Figure 7 caption: I suggest changing both instances of "in the rains" to "in precipitating downdrafts".
- Figure S1 caption: typo, "hPa".

---

## Author Comment (AC1)

Dear Reviewer,

Many thanks for your constructive comments on our manuscript. We will give point-to-point responses after finishing the comparisons of the model results presented in Figures 2 & 3 with the satellite observations. We have collected the monthly-averaged CO MLS satellite data for 2010-2020. However, the satellite data for $NH_3$ and $SO_2$ profiles in the UTLS are very limited.

The MIPAS/Envisat measured the UTLS $NH_3$ and $SO_2$ profiles during 2002-2011 as reported by Höpfner et al. (acp-13-10405-2013, acp-15-7017-2015, acp-16-14357-2016). But I am not sure whether the detection limits of MIPAS are low enough to validate our model results, especially for $SO_2$. Moreover, there are only two years of overlapped data between our model simulation and MIPAS observations. I will contact Dr. Höpfner for the MIPAS data.

It would be very appreciated if you could give more detailed suggestions on what $NH_3$ and $SO_2$ satellite data should be used for comparisons.

Sincerely,
Jianzhong Ma